

# Local-scale variability of snow density on Arctic sea ice

Joshua King[1], Stephen Howell[1], Mike Brady[1], Peter Toose[1], Chris Derksen[1], Christian Haas[2], Justin Beckers[3,4]

[1]Climate Research Division, Environment and Climate Change Canada, Toronto, M3H5T4, Canada
[2]Alfred Wegner Institute of Polar and Marine Research, Bremerhaven, Germany
[3]Department of Earth and Space Science and Engineering, York University, Toronto, Ontario, Canada
[4]Department of Earth and Atmospheric Sciences, University of Alberta, Edmonton, T6G2E3, Canada

*Correspondence to*: Joshua King (Joshua.King@Canada.ca)

**Abstract**

Local-scale variations in snow density and layering on Arctic sea ice were characterized using a combination of traditional snow pit and SnowMicroPen (SMP) measurements. In total, 14 sites were evaluated within the Canadian Arctic Archipelago and Arctic Ocean on both first (FYI) and multi-year (MYI) sea ice. Sites contained multiple snow pits with coincident SMP profiles as well as unidirectional SMP transects. An existing SMP density model was recalibrated using manual density cutter measurements (n=186) to identify best-fit parameters for the observed conditions. Cross-validation of the revised SMP model showed errors comparable to the expected baseline for manual density measurements (RMSE=34 kg m$^{-3}$ or 10.9%) and strong retrieval skill (R$^2$=0.78). The density model was then applied to SMP transect measurements to characterize variations at spatial scales of up to 100 m. A supervised classification trained on snow pit stratigraphy allowed separation of the SMP density estimates by layer-type. The resulting dataset contains 58,882 layer-classified estimates of snow density on sea ice representing 147 m of vertical variation and equivalent to more than 600 individual snow pits. An average bulk density of 310 kg m$^{-3}$ was estimated with clear separation between FYI and MYI environments. Lower densities on MYI (277 kg m$^{-3}$) corresponded with increased depth hoar composition (49.2%), in strong contrast to composition of the thin FYI snowpack (19.8%). Spatial auto-correlation analysis showed layered composition on FYI snowpack to persist over long distances while composition on MYI rapidly decorrelated at distances less than 16 m. Application of the SMP profiles to determine propagation bias in radar altimetry showed the potential errors of 0.5 cm when climatology is used over known snow density.

## 1 Introduction

The stratified nature of snow on sea ice provides a detailed history of interacting geophysical processes and synoptic-scale input. From its deposition on new ice, to melt in summer, these interactions are dynamic, leading to spatiotemporal heterogeneity at multiple scales. For large portions of the Arctic, early season cyclones drive rapid accumulation followed by sustained periods of cold air temperatures and high winds (Webster et al., 2018). The resulting snowpack is characteristically



shallow and subject to sustained temperature gradient metamorphism. Contrasting layers associated with these conditions, namely wind slab and depth hoar, form distinctive features of the winter snowpack (Sturm and Holmgren, 2002). Sequential precipitation and wind events contribute to layered complexity where mass is lost to open water (leads, polynyas), mixed-phase precipitation occurs (melt, ice features), or ice topography acts as an obstruction (drifts and dunes). Although structural

similarities exist at synoptic scales (e.g. Warren et. al., 1999), few studies have quantified local-scale variability on Arctic sea ice (10 m²; Iacozza and Barber, 1999; Sturm et al., 2002, Sturm et al., 2006) where layered snow strongly modulates optical and thermal properties at the surface (Ledley et al. 1991; Wu et al., 1999).

Accurate remote sensing observations of sea ice are dependent on spatially distributed knowledge of snow mass (a function of

thickness and density). For example, snow thickness represents a significant source of uncertainty in altimetry where isostasy is assumed for retrievals of sea ice thickness (Tilling et al., 2016; Kwok et al., 2019). Radar altimetry estimates of sea ice freeboard must also account for variations in snow density to determine an effective speed of propagation within the medium (Giles et al., 2007; Kwok et al., 2011). It is therefore of interest to develop objective representations of snow on sea ice to quantify potential errors and constrain models. *In situ* studies form the basis of these representations (Barber et al., 1995;

Warren et al., 1999; Sturm et al., 2002, Kwok and Haas, 2015) and are often extended spatially in model or satellite-based products (Kurtz et al., 2011; Lawrence et al., 2018; Liston, et al., 2018; Petty et. al., 2018). Given short length scales of variability, application of these approximations must be handled carefully where errors vary with vertical or horizontal resolution (Kern et al., 2015; King et al., 2015). Additionally, where basin scale inputs are required, recent changes to the Arctic climate system call into question how representative legacy snow climatologies are for current conditions (Kwok and

Cunningham, 2008; Laxon et al. 2013; Webster et al., 2014).

Although there is a need for enhanced representation of snow on sea ice, detailed *in situ* characterization can be challenging and costly to execute. Traditional snow pits are restricted to a single vertical dimension and require trained operators (i.e. Fierz et al., 2009). Adjacent snow pits or multiple profiles can be arranged to enhance horizontal dimensionality, but are cumbersome

to execute at scale (Benson and Sturm, 1993; Sturm and Benson, 2004). In many cases, a trade-off between horizontal and vertical resolution is necessary to balance available time with spatial coverage. Recently, penetrometer measurements with the SnowMicroPen (SMP; Schneebeli and Johnson, 1998) were used address this problem, providing a novel method to rapidly characterize snow structural properties including density (Proksch et al., 2015). The SMP provides mm-scale mechanical measurements which can be linked to vertical snow structure through modeling of the penetration process (Marshall and

Johnson, 2009; Löwe and van Herijnen, 2012). Taking less than a minute to complete a single vertical profile, there is potential to apply the SMP to snow on sea ice to provide detailed information for radiative transfer or mass balance applications at multiple scales.



In this study, we quantify local-scale variation of snow properties on Arctic sea ice using a combination of snow pit and SMP
measurements. SMP profiles are used to extend traditional snow pit analysis to characterize variations in density and
stratigraphy at spatial scales of up to 100 m. We validate the SMP density model of Proksch et al. (2015) at sites within the
Canadian Arctic Archipelago (CAA) and Arctic Ocean (AO) where adjacent density cutter profiles were collected on first-
and multi-year sea ice (FYI and MYI). The SMP density model is recalibrated using snow pits to identify best fit parameters
for the observed conditions. Traditional snow pit stratigraphy is then used to train a supervised SMP classifier, facilitating
evaluation of density by layer-type. The calibrated density model and layer-type classifier are applied to an independent set of
613 SMP profiles to discuss snowpack length scales of variability on Arctic sea ice. Finally, we apply the SMP derived
properties to an altimetry application, discussing propagation of errors as related to the observed snow structure.

## 2 Data and methods

### 2.1 Study areas and protocols

The measurements utilized in this study were acquired during two April field campaigns conducted near the time of maximum
snow thickness (Figure 1). The snow measurements coincided with NASA Operation IceBridge (OIB; 17 April 2016) and
ESA CryoVEx (26 April 2017) flights, aimed at improving understanding of inter-annual variability of Arctic snow and sea
ice properties. The snow measurements discussed here support local-scale analysis, fundamental to quantifying remote sensing
errors and linking physical processes at larger spatial scales.

The first measurement campaign took place near Eureka, Nunavut, Canada on landfast ice in the CAA (80.0°N 85.9°W; Figure
1a). Snow property measurements were collected over a 9-day period between 8 April 2016 and 17 April 2016 within Eureka
Sound and Slidre Fjord. Sea ice near Eureka was principally landfast FYI with embedded floes of MYI imported from the
Arctic Ocean via Nansen Sound. Similar to conditions reported in King et al. (2015), FYI near Eureka formed as large level
pans with limited deformation. Imported MYI was rough in comparison and heavily hummocked from exposure to previous
melt (Figure 1b). Mean ice thickness was 2.18±0.10 for FYI and 3.10±0.66 for MYI evaluated near Eureka (± *indicates
standard deviation)*. Measurements at Eureka were grouped by sites (250 x 100 m) with similar surface condition as determined
from visual inspection of RADARSAT-2 imagery (Figure 1b). At Eureka, a total of 8 sites were completed with 6 on FYI and
2 on MYI (Table 1). The average distance between sites was 15 km which were accessed via snowmobile from the Environment
and Climate Change Canada Eureka Weather Station.

A second campaign in April 2017 focused on the characterization of MYI in the Arctic Ocean (AO; Figure 1a). A Twin Otter
aircraft was used to access sites west of the Geographic North Pole from Alert, Nunavut, Canada along a CryoSat-2 track (see
Haas et. al, 2017). Measurements were carried out at 6 sites spanning 83.4°N and 86.3°N between 11 April 2017 and 13 April
2017 (Figure 1a). In contrast to the Eureka campaign, the AO sites traversed an extensive region of MYI with thickness





consistently greater than 3 m (Haas et. al, 2017). However, with limited time at each landing site, areas characterized were much smaller than at Eureka. The average distance between sites for Alert was 175 km, spanning a large gradient of ice conditions.

## 2.2 Snow pit measurements

The Eureka and AO campaigns had common goals to (1) collect adjacent snow pit and SMP profiles and (2) extend characterization from single, local snow pits to larger horizontal scales using SMP transects. As such, the common core of the measurement protocol were standard snow pits used as reference. For Eureka, an average of three snow pits were excavated per site (total n=20) and a single snow pit was completed per landing for the AO campaign (total n=6). Once excavated, stratigraphy was interpreted via visual inspection and finger hardness tests. Heights of the interpreted layers were marked on

the pit face and recorded. A 2-mm comparator card and 40x field microscope were used to classify each layer by standardized grain type as described in Fierz et al. (2009). Samples were then broadly categorized as rounded (integrating RGwp, RGxf grain types), faceted (FCso, FCsf), or depth hoar (DHcp, DHch), descriptive of predominate metamorphic processes. Trace amounts of recent snow were integrated into the rounded classification of some layers because of surface decomposition and wind rounded grain-type mixtures (i.e. DFbk).


Snow pit density was measured as continuous vertical profiles between the air-snow and snow-ice interfaces with a 100-cm$^3$ Taylor-LaChapelle cutter (75 g; Figure 2a). Extracted samples were weighed *in situ* with a shielded A&D EJ-4100 digital scale (±0.01 g accuracy). Measurements were rejected where the cutter could not be properly filled, such as in the presence of horizontal ice features or fragile microstructure. Previous studies have shown box-style cutter measurements to agree within

9% of high-certainty laboratory experiments (Proksch et al., 2016), however due to potential errors of omission from sample rejection, layer specific bias may be present.

## 2.3. SnowMicroPen (SMP) measurements

A single 4[th] generation SnowMicroPen (SMP) developed by Schneebeli and Johnson (1998) was used to measure profiles of penetration force ($F$). Operating at a constant speed of 20 mm s$^{-1}$, the high-resolution force transducer of the SMP was driven

vertically through the snowpack. The resulting profiles contained ~250 measurements of $F$ per millimeter, with a maximum $F$ of 45 N and resolution of 0.01 N. Given high surface hardness, a rigid metal mount was required to stabilize the sensor and prevent rebounding on initial penetration (Figure 2a). To preserve the penetrometer from impact with the ice surface, maximum penetration was set 1 cm shorter than the adjacent snow thickness. Coincident SMP profiles at were made at 26 snow pit locations to evaluate derived estimates of snow density (Table 1). Maintaining a horizontal separation of 10 cm, profiles were

located behind the snow pit wall in proximity to the manual cutter profile. After each profile, a snow depth probe was inserted into the SMP path to measure any unresolved thickness. The location of each profile was recorded with the GPS onboard the SMP (±5 m accuracy).





In addition to profiles at snow pit locations, SMP transects was established to characterize spatial variability (n=614; Table 1).
For Eureka, multi-scale sampling was applied where unidirectional sets of 10 profiles were separated at distances of 0.1, 1, and 10 m, in sequence (Figure 1c). Where time permitted, additional profiles were completed with 1 m spacing adjacent to the primary transect. An average of 69 SMP profiles were collected per site near Eureka (total n=550). It was not possible to execute an identical sampling strategy for AO sites due to time constraints. Instead, a single set of 10 profiles were spaced 1 m apart, parallel to the snow pit wall. An average of 11 profiles were made per site for the AO campaign (total n=63).

## 3. Snow density and layering on sea ice from SMP profiles

### 3.1 Basis for estimation of snow density from penetrometry

The SMP force signal, $F$, can be linked to physical properties of the snowpack though modeling of the penetration process, and related to density with an empirical model (Proksch et al, 2015). To do so, the fluctuating force signal measured by the SMP can be interpreted as the superposition of spatially uncorrelated ruptures and deflections (Marshall and Johnson, 2009). Conceptualised as a one-dimensional shot-noise process, Löwe and van Herwijnen (2012) reinterpret this relationship to compute estimates of microstructural length scale ($L$) without the need for *a priori* knowledge of snow structure. The derived quantity $L$ represents an idealized distance between two rupturing elements of the snow structure. Proksch et al. (2015; hereafter P15), building on the work of Pielmeier (2003), relate the microstructural property $L$ and median penetration force ($\tilde{F}$) to snow density through a bilinear regression using the following equation:

$$\rho_{\mathrm{smp}} = a + b ln(\tilde{F}) + c ln(\tilde{F})L + dL \quad (1)$$

where the coefficients (a, b, c, and d) were calibrated against micro-computed tomography (uCT) in alpine, Arctic, and Antarctic environments (Table 2). Detailed methodology regarding the two-point correlation function used to compute $L$ and other relevant parameters can be found in Löwe and van Herwijnen (2012).

### 3.2 Processing of SMP profiles for snow on sea ice

Prior to generating estimates of snow density, a series of pre-processing steps were applied to minimize SMP measurement uncertainty. First, profiles penetrating less than 90% of the measured snow thickness were removed from analysis to minimize vertical errors of omission. Force profiles were then evaluated to isolate signal artifacts by applying a minimum noise threshold of 0.01 N following Marshall and Johnson (2009). Signals below the threshold were removed and linear interpolation was applied to infill. Once filtered, $\tilde{F}$ and $L$ were computed following Löwe and van Herwijnen (2012), using a moving window of 5 mm with 50% overlap to meet an assumption of spatial homogeneity. Estimates of density ($\rho_{smp}$) were then calculated



by applying $\tilde{F}$ and $L$ in Eq. (1) with an appropriate set of coefficients. The resulting data are geo-located vertical profiles of $\rho_{smp}$ at 2.5 mm vertical resolution.

160

Small-scale lateral variations in stratigraphy made validation of $\rho_{smp}$ challenging where the reference snow pits were physically displaced. For example, pinching or expansion of layers from variations in ice topography at sub-meter scales could lead to large differences in compared density. Adapting an approach similar to Hagenmuller and Pilloix (2016), a matching process was applied to compensate for layered differences between the target and reference snowpack. To initiate the process, first-guess estimates of $\rho_{smp}$ were made with the P15 coefficients for profiles at snow pit locations. Derived profiles were then divided into arbitrary 5 cm layers and scaled randomly in thickness. Individual layers were allowed to erode or dilate by up to 75% of the original thickness, contributing towards a total permitted change of 10% per profile. A large number of scaling permutations were generated for each SMP profile using a brute force approach ($n = 1 \times 10^4$). Estimates of $\rho_{smp}$ were then extracted from the scaled profiles within the 3-cm height of the density cutter measurements and averaged. Best-fit alignment was selected where root mean square error (RMSE) was minimized between the thickness scaled $\rho_{smp}$ profiles and snow pit observed density.

Figure 3 shows an example of the matching process where a basal snow feature on MYI was poorly aligned between the first guess estimates of $\rho_{smp}$ and snow pit measurements. The profile was divided into 12 layers (Figure 3a) and scaled to identify best fit parameters for each layer (Figure 3b). An overall stretch of 6.5% was applied through the matching process, minimizing RMSE (54 kg m$^{-3}$) and improving correlation (R=0.83) in the example profile. Matching applied to all SMP profiles at snow pit locations resulted in a mean vertical absolute scaling of 7.4% or 1.7 cm.

### 3.3. Calibration of the SMP snow density model on sea ice

Once aligned, estimates of $\rho_{smp}$ were compared against *in situ* snow pit densities to quantify retrieval skill. An evaluation of the P15 parametrized density model is shown in Figure 4, including measurements from all 26 snow pit locations (n=196). Estimates of $\rho_{smp}$ were biased high relative to the snow pit reference with a large RMSE of 124 kg m$^{-3}$ (Table 2). The observed bias increased with density, leading to unrealistic overestimates for wind slab-classified samples (165 kg m$^{-3}$ RMSE). Conversely, errors were lowest for low-density depth hoar (96 kg m$^{-3}$ RMSE). Comparing estimates from the two campaigns, Eureka had a higher RMSE (135 kg m$^{-3}$) then measurements at AO sites (98 kg m$^{-3}$), but the discrepancy was related to lower overall density reported on MYI rather than campaign specific bias.

Despite a 41% RMSE, the P15 parametrized estimates of $\rho_{smp}$ were well correlated with snow pit measurements ($R^2$=0.72; p<0.01; Table 2). An ordinary least squares (OLS) regression was used to recalibrate P15 where snow pit measurements were available as reference. Values of $\tilde{F}$ and $L$ identified in best-fit scaling of the SMP profiles were used in the regression, along





with the corresponding density cutter measurements. A 10-fold cross-validation was applied where inputs were divided into roughly equal groups and used to train the regression with all but a single fold which was reserved for testing. Test-train permutations were iterated until each fold had been used independently in testing to minimize sampling bias. Coefficients were averaged across fold combinations and reported with RMSE and $R^2$ in Table 2.

Calibrated with the snow pit densities, retrieved coefficients of the regression differed substantially from P15 but remained identical in $R^2$ (Table 2). Applying the revised coefficients (referred to as K19a), RMSE was reduced to 41 kg m$^{-3}$ or 13% of the observed mean. Previously observed P15 bias was also minimized with no significant trend in residuals (0.1 kg m$^{-3}$). Given that the profiles used to evaluate model skill were physically displaced, it was unlikely that all matched comparisons were strong candidates for the recalibration. As such, a revised set of coefficients were prepared where outliers defined as the 95th

percentile of absolute error in the initial comparison were removed from the regression (>85 kg m$^{-3}$; n = 10). Data associated with these outliers were few in number and primarily associated with layer boundaries on FYI. Regression of the constrained input (K19b; Figure 4b) showed small differences in the retrieved coefficients (Table 2) along with improved skill ($R^2$=0.78; 34 kg m$^{-3}$).

To evaluate dependency of the regression parameters, their respective relationships with observed snow density are shown in Figure 5. Median force ($\tilde{F}$), once log-transformed, was well correlated with density in the combined ice surface and campaign dataset (R=0.76; Figure 1a). However, the observed relationship with $\tilde{F}$ weakened for samples collected on MYI (0.69 R), in particular those as part of the Eureka campaign (R=0.46). In contrast, the microstructural parameter $L$ remained well correlated with density regardless of ice type or campaign (R<-0.76; Figure 1b). In all coefficient parametrizations (P15 and K19) the

dependent variables and interaction term were found to be significant in the OLS regression. Although some dependency on snow conditions with respect to ice type were apparent, the K19b parametrization was applied globally as the relationship was unlikely to be driven by ice type but rather as some function of ice surface roughness which was unaccounted for in this study.

### 3.4 Classification of SMP density profiles by layer-type

To quantify the stratigraphic variability of snow density, a support vector machine (SVM) was implemented to partition the

SMP density profiles by layer-type (Cortes and Vapnik, 1995). SVMs apply hyperplanes in high-dimensional space to separate classes by maximizing distance from support vectors (i.e. a hyperplane which best delineates the nearest data pairs between classes). Automated learning methods have previously been applied to support classification of SMP profiles (e.g. Havens et al., 2012) and are well suited to rapidly process the available transect data. Learning methods for SVM classification were adapted from Scikit-learn (Pedregosa et al., 2011) and trained on snow pit (layer-type) and SMP ($\tilde{F}$, $L$, Penetration depth)

information extracted according the Sect. 3.2 procedures. Applying a linear kernel, the classifier was 10-fold cross-validated similar to the OLS regression, however, sampling was stratified to ensure a minimum of 10 samples per layer-type class in each fold. Classified estimates from each profile were smoothed with a median window to remove thin layers with thickness





less than 1 cm (window size=5). Cross-validated results of the SVM classifier were found have a prediction accuracy of 76% when evaluated against the snow pit reference samples.


Figure 6 shows an example of a classified SMP profile compared against snow pit observed stratigraphy on MYI near Eureka. Trained against a generalized layer-type classification scheme, the methodology is incapable of identifying inter-layer variations apparent in the manual snow pit observations. However, by identifying major transitions, the classified profile can be used to quantify differences in layered composition observed across both ice type and campaign. Moreover, by counting

the number of transitions between layer-type classifications an approximate count of snowpack layers can be made.

## 4 Variability of snow density and layering on sea ice

### 4.1 Snow pits

Central to the density model evaluation was the acquisition of a limited number of high-certainty snow pits (n=26). Each served as a density reference for the SMP calibration but also provided baseline information on stratigraphy to frame the transect

analysis. By the April timing of the campaigns, evidence of wind redistribution and temperature gradient metamorphism were widespread, characteristic of late winter Arctic snowpack. Mean thicknesses of the snow pits were 20.8±6.1 and 38.0±12.7 cm for FYI and MYI, respectively. Stratigraphic complexity was apparent on MYI where 7 layers were present on average as opposed to 4 on FYI. Bulk density of snow pits on FYI (320±33 kg m$^{-3}$) was higher in comparison to MYI (300±36 kg m$^{-3}$), a function of limited variation in ice surface topography on the level FYI near Eureka.


Consistently across snow pits, density was highest in proximity to the air-snow interface where rounded grain-types were prevalent. Commonly known as wind slab, these layers were a product of mechanical wind rounding and subsequent sintering. Corresponding grain classifications were mainly wind packed (RGwp) or faceted rounded (RGxf) types. Density of the wind slabs were comparable between ice environments and campaigns with an average of 375±49 kg m$^{-3}$ (Table 3). Lower density

slab features occurred where wind-broken precipitation (DFbk) was inter-mixed with the smaller mechanically rounded grains. For example, at Alert sites 6 and 8, ~3 cm of decomposing precipitation was present at the air-snow interface leading to lower surface densities (~222 kg m$^{-3}$). In general, these so-called soft slab features were more common on MYI where rough ice features buffered wind blown snow efficiently.

Layers below the surface slab were similar in appearance but inspection of the grains revealed tightly packed facets. Distinct in microstructure, these former wind slabs showed clear signs of kinetic growth while maintaining a well bonded structure. Density of the mid-pack faceted layers were comparable to the surface features for the AO campaign (380 kg m$^{-3}$) but were slightly lower for Eureka (287 kg m$^{-3}$). At the base of the snowpack were multiple layers of large diameter depth hoar, texturally distinct from the overlying slab and faceted layers. Microstructure of the depth hoar was characterized by weakly bonded cups



(DHcp) at times clustered as large chained units (DHch). The unconsolidated structure of the depth hoar was fragile, often collapsing when inspected by touch or tool. Density of the depth hoar layer was consistently lowest, with a small range of observed variability (Table 3).

## 4.2 SMP profiles

Despite layered similarity, it was difficult from snow pits alone to directly compare proportional composition or determine

process scales over which structural correlations might persist. The SMP transects presented a unique opportunity to extend analysis beyond the point-scale and link layers between snow pits by drastically increasing the number and spatial diversity of profiles. Once processed, the SMP profiles collected on sea ice provided 58,882 estimates of snow density, representing approximately 147 m of vertical variation. Each profile also contained estimates of proportional composition by layer-type though automated classification, facilitating layer density comparisons and spatial analysis.


Figure 7 shows the aggregated results of the SMP transect profiles where snow density measurements were separated by ice surface and layer type. Separated by ice type (FYI and MYI), bulk densities (vertically integrated) are represented in two overlapping but distinct distributions (Figure 7a). Profiles collected on FYI, and therefore exclusively near Eureka, formed a negatively skewed distribution with mean density of $327 \pm 42$ kg m$^{-3}$ (n=402). In contrast, densities on MYI were positively

skewed with a mean of $277 \pm 30$ kg m$^{-3}$ (n=211). Separating the MYI profiles by campaign shows small differences between Eureka ($272 \pm 27$ kg m$^{-3}$, n = 148) and the AO ($290 \pm 31$ kg m$^{-3}$, n=63). However, these difference were small in comparison to the larger shift between MYI and FYI. Overall, the opposing skew shows a clear shift in snow density by ice surface type for the observed domains independent of campaign.

Given that all profiles were classified by layer-type (see Sect. 3.3), it was also possible to evaluate distributions of density separately for rounded, faceted, and depth hoar features (Figure 8; Table 4). Composition of the thin Eureka FYI snowpack ($18.1 \pm 8.8$ cm) was primarily faceted, represented by $50.0 \pm 18.1\%$ of total thickness on average. Measurements classified as faceted had on average a density of $336 \pm 43$ kg m$^{-3}$, forming a negatively skewed distribution (Figure 8c). Rounded layers in proximity to the air-snow interface were thinner, accounting for $30.2 \pm 15.0\%$ of total thickness, with a mean density of $352 \pm 51$

kg m$^{-3}$, also represented in a negatively skewed distribution (Figure 8). At the base of the snowpack on FYI, depth hoar was both the smallest fractional component ($19.8 \pm 18.4\%$; Figure 8) and had the lowest mean density ($248 \pm 27$ kg m$^{-3}$; Figure 7d).

Composition on MYI shifted towards a larger proportion of depth hoar ($49.2 \pm 16.3\%$) within the overall thicker snowpack ($34.7 \pm 16.8$ cm). Density of the depth hoar on MYI was comparable to FYI with means of $247 \pm 15$ and $257 \pm 18$ kg m$^{-3}$ for the

Eureka and AO sites, respectively. Mid-pack faceted layers were well represented at $35.5 \pm 13.7\%$, albeit with lower density overall at $301 \pm 43$ kg m$^{-3}$ and notable decreases at Eureka ($294 \pm 41$ kg m$^{-3}$). Layers classified as rounded composed the remainder of the volume at $15.3 \pm 8.5\%$. The fractional distribution of rounded layers on MYI was bimodal (Figure 8),





corresponding with the presence of decomposing precipitation in mixed-type layers. Mean density of rounded layers on MYI were on average 291±65 kg m$^{-3}$ with slightly higher densities observed at AO sites (306±67 kg m$^{-3}$) over Eureka (285±63 kg m$^{-3}$).

As a proxy for the number of observed layers, transitions between layer-type classifications were summed for each profile. Figure 9 shows probability densities associated with layer-type transitions separated by ice type. As in the snow pits, stratigraphic complexity was greater on MYI with an average of 6.9 layer-type transitions as opposed to 4.5 on for FYI. Of the transitions, those associated with faceted layers on MYI were most prevalent with an average of 2.9 per profile. Faceted transitions on FYI were fewer in number at 1.9 on average. Depth hoar layers were 2.8 on average for MYI and 1.5 for FYI. Finally, in both environments the average number of rounded transitions was 1.1 with no more than 3 transitions identified in any profile.

## 4.3 Length scales of variability

Large standard deviation relative to most layer-type fractions indicated strong stratigraphic variability within the SMP transect dataset. Given that these variations are driven at local-scales by ice topography and weather (wind and precipitation), it can be expected that structural similarities persist at some process scale. To evaluate differences in length scale, estimates of spatial auto-correlation for layer-type composition were computed using Moran's I (Moran, 1950):

$$I(d) = \frac{\frac{1}{w}\sum_{i=1}^{n}\sum_{j=1}^{n}w_{ij}(x_i - \bar{x})(x_j - \bar{x})}{\frac{1}{n}\sum_{i=1}^{n}(x_i - \bar{x})} \quad (2)$$

where $x$ are rounded, faceted or depth hoar volume fractions at locations $i$ and $j$ displaced at a lag distance of $d$. Equation (2) was evaluated for pairs of profiles separated by $d$ from 1 to 100 m in 1 m increments for each layer-type. To compensate for geo-location errors, a tolerance of ±5 m was applied to $d$, corresponding with GPS accuracy of the SMP. Weighting ($w$) of Eq. (2) takes on a value of 1 when pairs were displaced at $d \pm 5$ m and 0 otherwise. On average, 412 pairs were evaluated per lag of $d$ on FYI and 333 on MYI. The resulting spatial auto-correlation analysis of snowpack fractional composition is presented in Figure 10. At scales beyond 100 m the number of profile pairs were limited given the ~250 m length of each site and are therefore not presented.





On FYI, all layer-types showed similar trending in auto-correlation with distinct minimums spaced at approximately 40 m
        (Figure 10). After an initial decline, depth hoar fraction on FYI remained moderately correlated at scales of 100 m and also
        was the strongest of the layer-type correlations (R=0.70 at 52 m). While the magnitude of the remaining rounded and faceted
        fraction layer-type correlations were lower, spatial trends on FYI were highly correlated with each other. In contrast,
        correlations dropped quickly on MYI for all layer-types and remain low at scales of 100 m (Figure 10). Faceted-type layer
fraction maintained spatial correlation for the longest period on MYI, but reached 0 at only 16 m. The result suggests spatial
        persistence of layered features on FYI beyond the available data. In contrast, strong variations in snow structure on MYI at
        short scales indicate the presence of distinct drivers and variability absent on the level FYI near Eureka.

**5. Implications for radar propagation in snow on sea ice**

        In the context of radar altimetry and sea ice freeboard retrievals, errors related to snow density can be described as a propagation
bias where speed of the interacting wave is reduced in snow (Kwok et al., 2011). Without accounting for this reduction, radar
        measured distance to the ice surface may be overestimated, or in a retrieval, underestimate the height of the sea ice freeboard.
        Established empirical relationships with permittivity (i.e. Ulaby et al., 1986) can be leveraged to quantify reductions in wave
        speed ($c_s$):

$$c_s = c(1 + 0.51\, \rho_s\,)^{-3/2} \quad (3)$$


        where $\rho_s$ is observed or modeled snow density. Estimates of path length difference ($\delta_p$) relative to propagation in free space
        can then be computed with respect to variations in snow thickness ($h_s$):

$$\delta_p = h_s \left( \frac{c}{c_s} - 1 \right) \quad (4)$$

Utilizing these equations, differences in propagation bias were evaluated for two scenarios where snow density was (1)
        determined from climatology and (2) parametrized from SMP measurements. The first configuration mirrors common practice
        in radar altimetry where the two-dimensional quadratic of Warren et al. (1999) was used to compute bulk density based on
        location and month. The second configuration leverages the high vertical resolution of the SMP profiles to approximate
        variations in wave propagation at the mm-scale. One-way path length difference from the SMP ($\delta_p^{\mathrm{SMP}}$) was calculated as the
summation of $\delta_p$ for each 2.5 mm vertical estimate. In both scenarios, height of the snowpack ($h_s$) was defined as SMP total
        penetration and $\delta_p$ was evaluated at each profile location.





Given the small geographic extent of the Eureka campaign, climatological estimates of snow density predicted no spatial variability with a mean of 298 kg m$^{-3}$. This was unsurprising given that the Warren et al. (1999) estimates are generally considered invalid within the CAA due to a lack of observations in the region. As such, a static value of 300 kg m$^{-3}$ was applied, representative of a typical static parametrization used in altimetry studies (e.g. Tilling et al., 2016). For AO sites, climatological estimates fell within a narrow range between 317 to 321 kg m$^{-3}$, across a nearly 3° difference in latitude. From climatology, the average reduction in wave speed relative to free space $(\overline{c_s/c})$ was estimated to be 0.809 for Eureka and 0.799 for AO sites.. With little variability in density, $\delta_p$ was driven strongly by snow thickness, where longer physical paths lead to larger delays (Figure 11a). As a result, predicted bias was greatest on MYI North of Alert (9.2 cm), despite slower expected propagation within the higher-density Eureka snowpack.

For the second scenario, average wave speeds relative to free space remained similar for Eureka (0.803) but were increased for AO profiles (0.815).Estimates of propagation bias computed with explicit representation of density ($\delta_p^{\mathrm{SMP}}$) showed limited sensitivity to the observed variations (Figure 11a). At $h_s$ corresponding with the Eureka FYI mean (18.2±1 cm) the spread in $\delta_p^{\mathrm{SMP}}$ was approximately 2.3 cm with an average propagation bias of 4.7 cm. Conversely, for AO sites (39.7±1 cm) the spread in $\delta_p^{\mathrm{SMP}}$ increased to 3.8 cm with a mean propagation bias of 9.3 cm. To quantify errors related to the use of climatology over *in situ* observations, the two scenarios were differenced ($\delta_p^{\mathrm{w99}}$- $\delta_p^{\mathrm{SMP}}$; Figure 11b). Assuming $\delta_p^{\mathrm{SMP}}$ as truth, potential errors on FYI ranged from -4.0 to 2.6 cm with a mean of 0.5±0.6 cm. On MYI, where densities were generally lower, the mean difference was -0.5±0.8 cm, spanning a small range of -2.5 to 2.3 cm. Errors on MYI between the AO and Eureka campaigns were in close agreement with means of -0.5±0.7 and -0.2±0.9 cm, respectively.

## 6. Discussion

Considerable skill was demonstrated in SMP retrievals of snow density by Proksch et al. (2015) but there had been no previous application on sea ice or in environments with a comparable snowpack dominated by wind slab and depth hoar. Evaluation of the P15 coefficients at 26 snow pits on sea ice showed a strong positive bias in SMP derived density compared to manual density cutter measurements. As a technical limitation, Proksch et al. (2015) noted that future hardware revision of the SMP would necessitate recalibration due to differences in signal digitization. The P15 results appear to confirm this limitation which was addressed with an OLS regression of the coincident density cutter measurements and SMP profiles. The recalibrated coefficients showed errors 10.9% of the observed mean when evaluated across a selected set of reference measurements (34 kg m$^{-3}$, n=185). This was comparable to the P15 reported error (10.6%), and within the range of the reported skill (R$^2$=0.64-0.80). Differences in error between ice environments and campaigns were nominal (<4 kg m$^{-3}$), demonstrating confidence in the application of a globally optimized set of coefficients.



Observed errors in the snow density may be accounted for by limitations in the snow pit and SMP procedures. First, density

cutter measurements used as reference, as opposed to high-certainty micro-CT, include baseline errors of up to 8% (Proksch et al., 2016). Sampling in depth hoar was a particular challenge where insertion of the cutter lead to collapse of the fragile microstructure. Evaluation of depth hoar was further complicated by low signal-to-noise where weakly bonded grains produced little variation in the SMP measured force. As a result, errors associated with depth hoar (14.0%) were greater than rounded (10.7%) or faceted (8.6%) layers. Errors associated with the higher density slab features may also be present due to

unaccounted interactions between failing elements in the penetration model (Löwe and van Herwijnen, 2012). Representing two extremes, wind slab and depth hoar presented challenging retrieval scenarios, however the errors appears consistent with those expected from manual density cutter measurement and previous study.

Differences between the two ice type environments (MYI and FYI) showed median force ($F$) to be an unreliable predictor of

snow density, particularly on MYI (Figure 5). This was consistent with Marshall and Johnson (2009) who first identified environment specific sensitivity of the SMP force leading Proksch et. al. (2015) to include the microstructural term $L$ in the empirical model. The relationship between $L$ and snow density was found to be independent of ice environment or campaign, acting to balance the retrieved density where signal-to-noise was poor. Limiting the regression inputs in Sect. 3.3 to profiles collected only on MYI shows the strong influence of $L$ where retrieved coefficients place increased weight on microstructure

(+42%). The observed dependencies were unlikely to be driven by ice type but rather by associated differences in ice topography and therefore retained snow structure. Differences in snow structure were clear between the two ice environments, however quantitative evaluation of how ice topography might be used to further refine coefficients in Eq. (1) was beyond the scope of this work and will require measurements in deformed FYI environments.

An average bulk density of 310±37 kg m$^{-3}$ was measured across all profiles included in this study (n=615). Separated by ice type, contrasting distributions were presented where density on the wind swept FYI was typically greater (Figure 7a). Despite separation by a full year, and hundreds of kilometers, the MYI measurements from the AO and Eureka sites were similar in bulk density at 289 and 272 kg m$^{-3}$, respectively. Evaluation of the snowpack structure showed depth hoar composition to be a driver of reduced density on MYI relative to FYI. At the local scales, melt ponds and hummocks on MYI serve to trap larger

amounts of snow earlier in the season (Radionov et al., 1996, Sturm et al., 2002). Coupled with strong temperature gradients, favourable conditions for the development of a substantial depth hoar layer were common to most MYI sites. Ice topography control on the internal structure of the snowpack, and ultimately bulk density, was also apparent in the length scale analysis (Figure 10). Rapid decorrelation of layered structure on MYI suggested that the hypothesised ice topography interactions occurred on relatively short length scales, driving high spatial variability particularly in depth hoar. In contrast, where large

smooth floe were typical for Eureka, covariance of the layer composition on FYI persisted over large distances (>100 m). The observed variations showed a periodicity common to length scales associated with snow dunes or interactions between drifted elements (Sturm et. al, 2002; Moon et al., 2019). In the future it would be instructive to evaluate how information on surface



roughness can be used to constrain understanding of internal snowpack structure between ice type environments where clear contrast exists.


A limited number of studies were available to place the observed stratigraphy in the context of other Arctic regions. Measurements collected during N-ICE2015 (Merkouriadi et al., 2016) in the Atlantic sector of the Arctic Ocean described a predominantly faceted composition on FYI (48%) and second-year ice (54%). Although the data presented in this study contains no second year ice, the FYI faceted composition is in agreement, as are bulk densities in the relatively flat and wind

swept environments. Faceted layers observed in both campaigns originated as surface slabs, buried by successive winter storms. With density and hardness comparable to the overlying wind slab, the differentiating factor in evolution was length of exposure to strong temperature gradients (Derksen et. al, 2009; Domine et. al., 2012). Snow pits consistently had measured temperature gradients sufficient for kinetic grain growth on FYI (25.7 C m$^{-1}$ on average; Colbeck 1983). While these layers had not fully converted to depth hoar due to high initial density (Akitaya, 1974), the larger faceted crystals were texturally

distinct, separating the wind slab from depth hoar. Improved understanding of how these faceted layers evolve at larger scales may be important in remote sensing or thermodynamic applications as they contribute enhanced scattering and reduced thermal conductivity relative to their wind slab origin types.

Snow stratigraphy reported during SHEBA in the Beaufort Sea (Sturm et al., 2002) showed greater wind slab composition

overall (42%), although this varied considerably with ice surface roughness. Analysis during SHEBA suggested that increased wind slab fraction was associated with smoother ice classes and therefore thinner snowpack. Similar observations inferred from the SMP profiles showed fractional composition by wind slab to increase by 46% on FYI over MYI because surface roughness conditions were much smoother. At small-scales, portions of slab and hoar approached parity during SHEBA where precipitation was intercepted earlier and retained within hummocks. Consolidating the wind slab and faceted layer

classifications on MYI, the SMP derived composition was also roughly equal when compared to depth hoar (52% for AO and 51% for Eureka). In the case of both N-ICE2015 and SHEBA, direct comparison of the snowpack composition is difficult due to different measurement protocols, but common themes regarding the influence of ice topography on snowpack stratigraphy were clear.

Applying the SMP measurements to compute differences in radar propagation showed small differences compared the use of climatological density. In comparison to the Warren et al. (1999) climatology, density on MYI in the AO was approximately 10% lower, insufficient to drive strong uncertainty in radar derived freeboard. Amongst the SMP derived parameters, penetration bias most greatly influenced by fractional composition by rounded-type layers (R=-0.73) on both FYI and MYI. High surface densities resulted in wave speeds 3% slower than the remaining snowpack, having the most significant impact

with respect to increased proportion. However, the overwhelming influence on the propagation bias remains snowpack thickness (R=0.97).





Although the result regarding density uncertainty in radar altimetry suggests that use of constants to represent density may be sufficient, several issues persist that should be addressed prior to making a conclusion. First, the work of Nandan et al. (2017) demonstrated that penetration is negatively impacted by the presence of brine within the snow volume on FYI. In section 5, the role of salinity was not addressed for profiles on FYI and is likely to be a much larger and inverse uncertainty. Second, in addition to density, Proksch et al. (2015) demonstrated the ability to retrieve snow microstructural properties from SMP signals. Lacking a reference for calibration, microstructural quantities were not evaluated as part of this study but in the future could be used to address waveform uncertainty corrected in some products (Ricker et al., 2014).

## 7. Conclusions

The recent shift towards younger Arctic sea ice (Maslanik et al., 2011) along with increased winter precipitation (Zhang et al., 2019) are likely to drive variations in snow structure whose full characterization will require a combined *in situ*, model, and remote sensing framework. Combinations of lidar and radar altimetry have been used to estimate snow depth on sea ice from space but no remote sensing methods are yet available to directly characterize density (Kwok and Markus, 2018; Lawrence et al., 2018). As such, *in situ* and model support are critical to fully address spatiotemporal variations in snowpack properties including mass and permittivity. The ability to rapidly collect SMP profiles across a broad set of features is attractive in this regard. Where a single snow pit represents only a snapshot of potential configuration, multiple SMP profiles can be leveraged to generate snow property distributions. Such data are highly desirable as they minimize or remove entirely, subjective bias introduced by operator decision or skill. This may allow meaningful parametrization of models with the intention of transferring local-scale campaign based analysis to larger domains. Although application of the SMP on sea ice shows great potential to meet this need, it does not replace standard snow pit methods required to frame more specific or larger-scale analysis.

The SMP transect analysis demonstrated contrasting controls on snow density and layering specific to ice type and depositional environment. While snowpack structure of level FYI appears to persist at scales beyond 100 m, MYI was highly variable at distances beyond 20 m. The role of ice topography and snow thickness variations as hypothesized drivers of these differences should be studied in further detail to assess if variations in snowpack structure can be inferred from knowledge of the ice surface itself. Such information would be valuable for remote sensing of sea ice studies where snowpack variations stand as a critical uncertainty.

The spatially distributed mm-scale estimates of snow density and layering introduced in this study provide novel information on multi-scale variability in Arctic sea ice covered domains. We hope these measurements will be relevant for applications beyond the altimetry case study discussed here such as in model development (Petty et al., 2018; Liston et al., 2018; Landy et



al., 2019) and to assist in the definition of future satellite candidate missions including the Copernicus polaR Ice and Snow
Topography Altimeter (CRISTAL).

**Code and data availability**

Code and data to reproduce all figures and analysis are available at https://github.com/kingjml/SMP-Sea-Ice (DOI pending).

**Author contributions**

J.K. wrote the manuscript with input from all authors. J.K. and S.H. designed the experiment. J.K. and M.B. preformed the
analysis. J.K, S.H., C.H., C.D., and J.B., collected the field measurements.

**Competing interests**

The authors declare that they have no conflict of interest.

**Acknowledgements**

We thank Polar Continental Shelf Program, Eureka Weather Station, Canadian Forces Station Alert, and Defence Research
and Development Canada for their assistance and logistical support of this work. Financial support for fieldwork was provided
by Environment and Climate Change Canada (ECCC), European Space Agency (ESA), Natural Sciences and Engineering
Research Council (NSERC) and Canada Research Chair (CRC) programs, and Alfred Wegener Institute for Polar and Marine
Research (AWI). We thank the legendary Arvids Silis, as well as Tom Newman, Jack Landy, Julienne Stroeve, Jeremy
Wilkinson, Jim Milne, Chris Brown, Andrew Platt, and Troy McKerral for their dedicated efforts to make the Arctic Ocean
and Eureka campaigns a success. Analysis in this study uses Python open source scientific computing packages including
Pandas, NumPy, scikit-learn from the SciPy ecosystem.

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



**Tables**

**Table 1: Summary of measurements completed as part of the Eureka (E) and Arctic Ocean (A) campaigns.**

| Site Details | | | Location [DD] | | Measurements [#] | |
|---|---|---|---|---|---|---|
| *ID* | *Type* | *Date* | *Latitude* | *Longitude* | *Pits* | *SMP Profiles* |
| A2 | MYI | 11/04/2017 | 83.9834 | -66.3509 | 1 | 12 |
| A3 | MYI | 11/04/2017 | 83.4421 | -64.4156 | 1 | 12 |
| A5 | MYI | 13/04/2017 | 84.8578 | -69.7044 | 1 | 12 |
| A6 | MYI | 13/04/2017 | 85.4446 | -73.4211 | 1 | 12 |
| A7 | MYI | 12/04/2017 | 83.4421 | -64.4154 | 1 | 4 |
| A8 | MYI | 12/04/2017 | 86.1987 | -79.3859 | 1 | 12 |
| E1 | FYI | 08/04/2016 | 79.9629 | -86.0019 | 3 | 31 |
| E2 | FYI | 09/04/2016 | 79.9944 | -86.4462 | 3 | 41 |
| E3 | FYI | 10/04/2016 | 80.0785 | -86.7794 | 3 | 70 |
| E4 | FYI | 11/04/2016 | 79.8440 | -86.8051 | 3 | 70 |
| E5 | MYI | 13/04/2016 | 79.9829 | -86.2933 | 3 | 85 |
| E6 | FYI | 14/04/2016 | 80.0211 | -86.7856 | 3 | 92 |
| E7 | FYI | 15/04/2016 | 79.9716 | -86.7909 | 3 | 100 |
| E8 | MYI | 17/05/2016 | 79.8135 | -86.8083 | 2 | 63 |

**Table 2: Model coefficients for Eqn. (1) as originally defined in Proksch et al. (2015; P15) and recalibrated as part of this study to estimate snow density sea ice (K19).**

| Set | Samples [#] | Regression Coefficients | | | | Metrics | |
|---|---|---|---|---|---|---|---|
| | | $a$ (kg m$^{-3}$) | $b$ (N$^{-1}$) | $c$ (N$^{-1}$ mm$^{-1}$) | $d$ (mm$^{-1}$) | *RMSE* [kg m$^{-3}$] | $R^2$ |
| P15 | 196 | 420.47 | 102.47 | -121.15 | -169.96 | 130 | 0.72 |
| K19a | 196 | 315.61 | 46.94 | -43.94 | -88.15 | 41 | 0.72 |
| K19b | 186 | 312.54 | 50.27 | -50.26 | -85.35 | 34 | 0.78 |

**Table 3: Mean density cutter measurements from snow pits separated by layer type and campaign.**

| Campaign | Pits [#] | | Density FYI [kg m$^{-3}$ (#)] | | | Density MYI [kg m$^{-3}$ (#)] | | |
|---|---|---|---|---|---|---|---|---|
| | *FYI* | *MYI* | *Rounded* | *Faceted* | *Depth hoar* | *Rounded* | *Faceted* | *Depth hoar* |
| Eureka | 17 | 3 | 375 (19) | 357 (46) | 232 (36) | 381 (6) | 287 (24) | 218 (10) |
| Arctic Ocean | 0 | 6 | - (0) | - (0) | - (0) | 373 (4) | 380 (18) | 275 (38) |
| Combined | 17 | 9 | 375 (19) | 357 (46) | 232 (36) | 375 (10) | 343 (42) | 249 (48) |




**Table 4: Snow density and layering derived from SMP profiles at each field campaign site using Eqn. (1) and the automated classification procedures described in Sect. 3.4.**

| Site | | SMP derived properties | | SMP derived composition | | |
| --- | --- | --- | --- | --- | --- | --- |
| *ID* | *Type* | *Penetration* | *Density* | *Rounded* | *Faceted* | *Hoar* |
| | | [cm] | [kg m$^{-3}$] | [%] | [%] | [%] |
| A2 | MYI | 44.2 | 289 | 11.3 | 36.8 | 51.9 |
| A3 | MYI | 33.6 | 302 | 20.3 | 42.5 | 37.2 |
| A5 | MYI | 39.2 | 326 | 11.8 | 60.8 | 27.4 |
| A6 | MYI | 34.5 | 267 | 14.0 | 29.1 | 56.9 |
| A7 | MYI | 34.7 | 279 | 12.7 | 14.8 | 72.5 |
| A8 | MYI | 45.4 | 272 | 8.2 | 20.6 | 71.2 |
| E1 | FYI | 13.8 | 364 | 38.2 | 53.7 | 8.1 |
| E2 | FYI | 17.3 | 346 | 31.6 | 59.0 | 9.4 |
| E3 | FYI | 15.9 | 342 | 39.1 | 52.1 | 8.8 |
| E4 | FYI | 20.2 | 285 | 23.3 | 33.1 | 43.6 |
| E5 | MYI | 32.5 | 279 | 15.4 | 36.2 | 48.4 |
| E6 | FYI | 15.4 | 336 | 34.1 | 57.6 | 8.3 |
| E7 | FYI | 22.4 | 309 | 22.3 | 48.5 | 29.3 |
| E8 | MYI | 32.4 | 268 | 17.4 | 33.1 | 49.5 |



## Figures

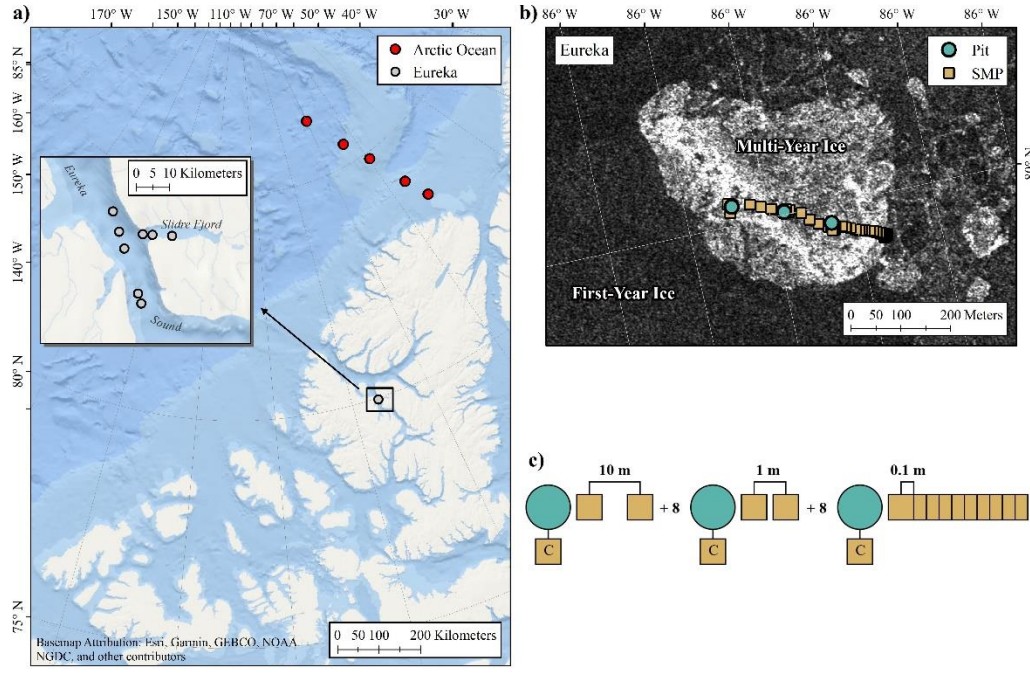


**Figure 1: Overview of the Eureka and Arctic Ocean (AO) snow on sea ice campaigns (a). Unidirectional SnowMicoPen (SMP) transects were collected at multiple sites to evaluate spatial variability of snowpack properties (b; Eureka MYI site shown) with sets of 10 profiles separated at distances of 0.1, 1, and 10 m, in sequence (c). Co-located SMP profiles were collected at all snow pit locations to calibrate the SMP density model of Proksch et al. (2015). Background of (b) shows RADARSAT-2 imagery near Eureka**
**where bight returns indicate rough multi-year ice. RADARSAT-2 Data and Products © MacDonald, Dettwiler and Associates Ltd. (2019). All Rights Reserved. RADARSAT is an official trademark of the Canadian Space Agency.**

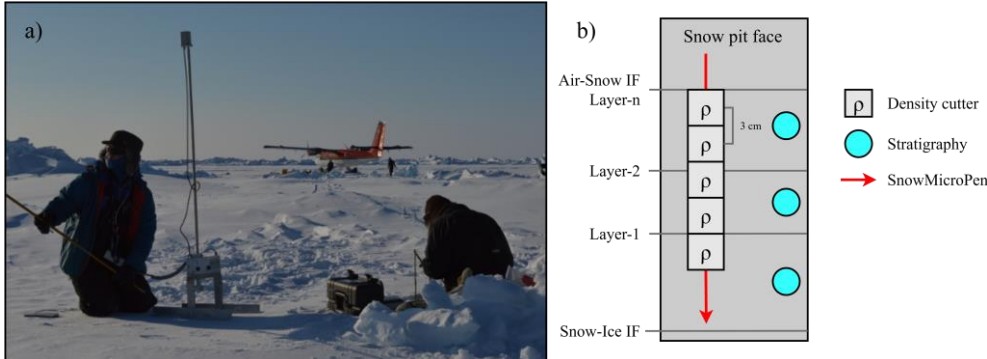

**Figure 2: Photo of typical SMP and snow pit measurements on MYI (a) and sketch of the sampling procedure (b). Manual density**
**measurements (3-cm height) were collected as continuous profiles between the air-snow and snow-ice interfaces (IF). SMP profiles were compiled 10 cm behind the pit face at each site. A rigid mount was used to stabilize the SMP while penetrating hard surface slabs (a).**

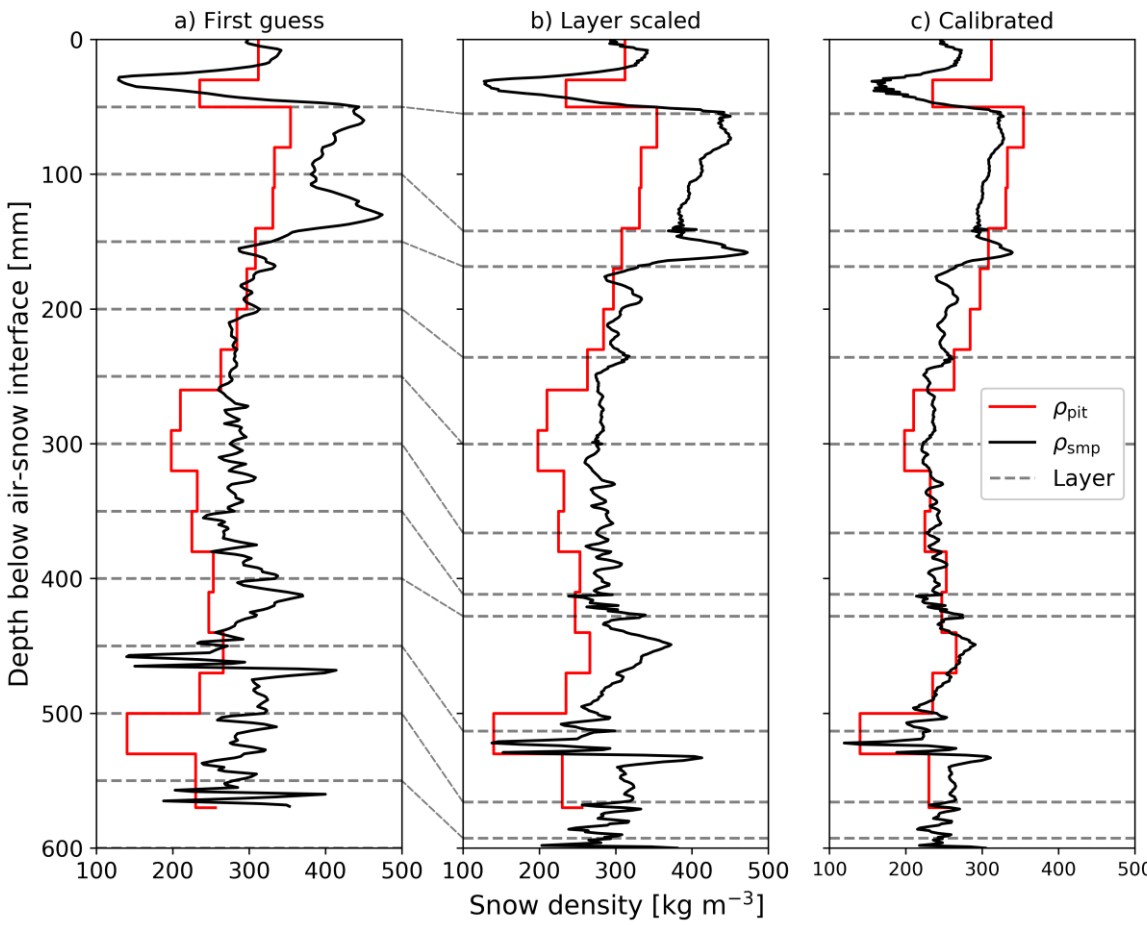


**Figure 3: SMP processing steps where first guess estimates of $\rho_{smp}$ (a; black lines) are used to improve alignment with snow pit measurements of density (b; red lines) prior to recalibration and computation of final estimates (c). To begin alignment SMP profiles are divided into arbitrary 5 cm layers (dotted lines) and scaled randomly in thickness. Best fit alignment is selected where RMSE between the SMP estimates and snow pit measurements are minimized. The matching process accounts for differences in the target**
**snowpack due to the 10 cm separation between profiles. The example shown is for Eureka site 5 on MYI.**

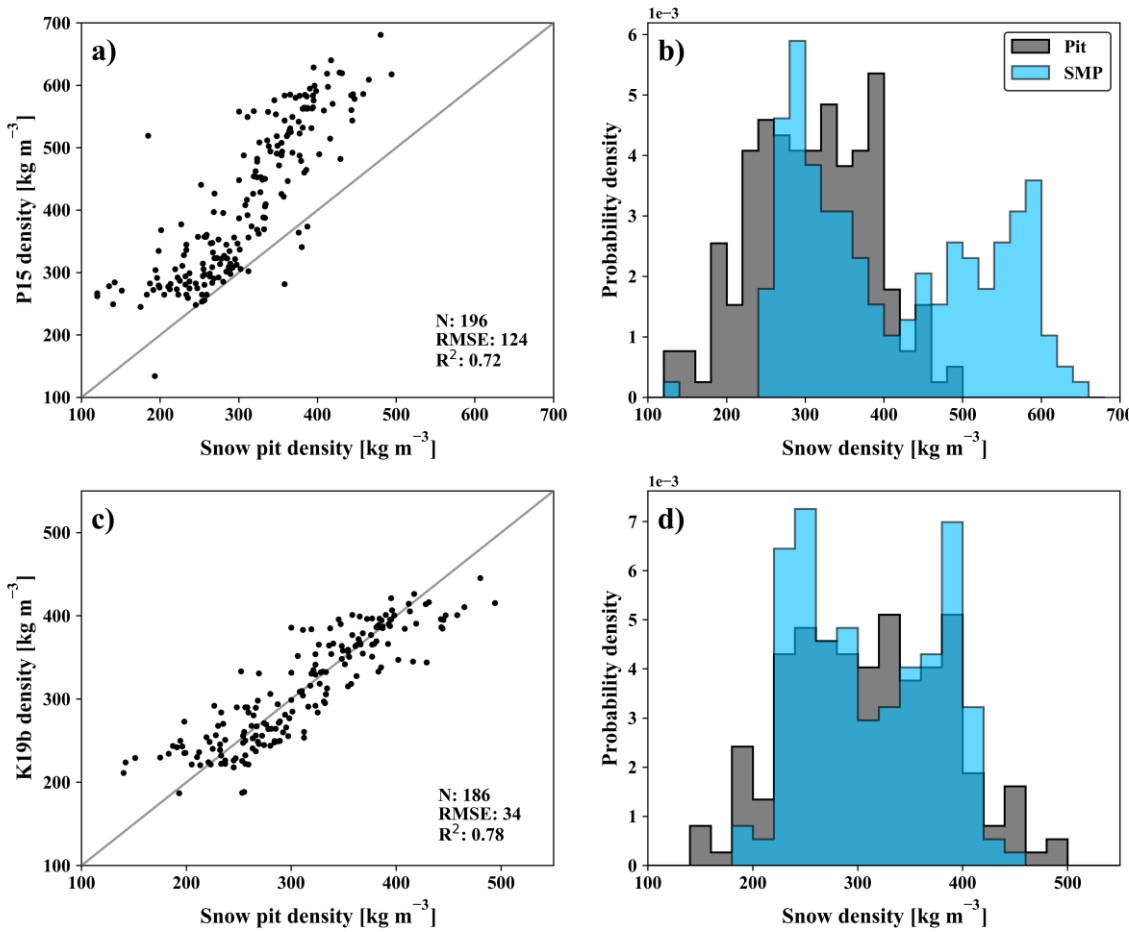

**Figure 4: Evaluation of the original SMP density model of Proksch et al. (2015) (P15; a) and recalibrated coefficients for snow on sea ice (K19b; b). Retrieved distributions are shown for the P15 (b) and K19b (d) parameterizations of Eqn. 1 with a common bin size of 20 kg m⁻³. In all cases, the reference measurements are manual density cutter measurements of snow density.**






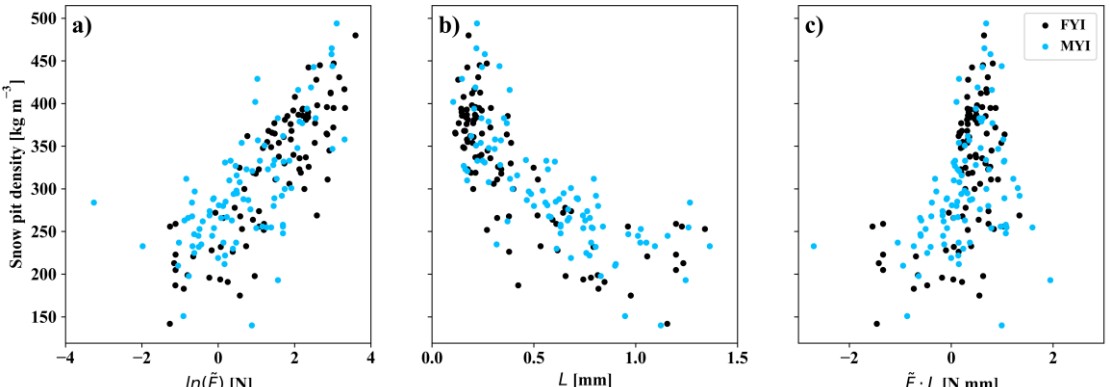

**Figure 5: Comparison of the SMP regression parameters and corresponding snow pit observed density. Parameters include log-transformed median force ($ln(\tilde{F})$, a), microstructure length scale ($L$, b) and an interaction term ($\tilde{f}L$, c). Relationships are separated by ice type (FYI and MYI).**


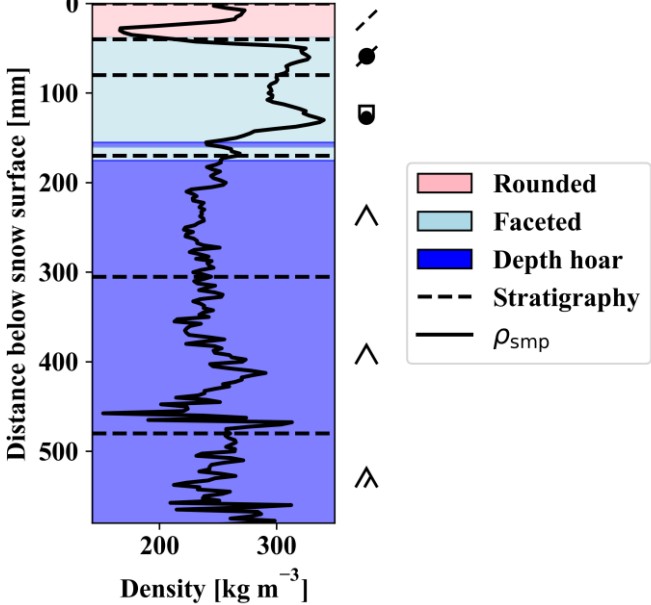

**Figure 6: Automated layer-type classification of a SMP profile collected on sea ice where colours indicate classification result. Horizontal dashed lines indicate heights of snow pit observed stratigraphic layers at the same location. Snow layer classification follows standardized colours and symbols described in Fierz et al., (2009).**






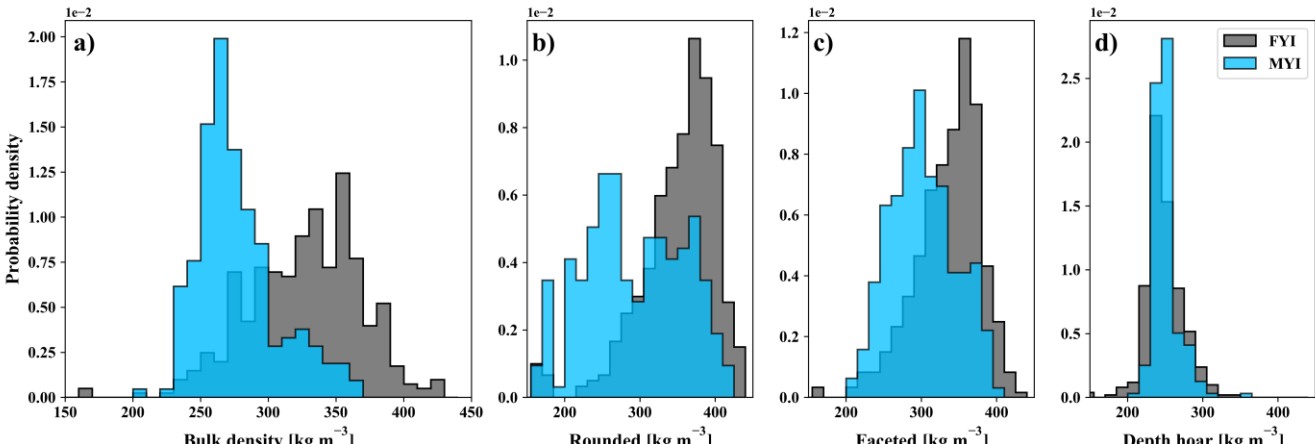

**Figure 7: Bulk density (vertically integrated) derived from SMP profiles on first year (FYI, n = 402) and multiyear (MYI, n = 211) sea ice (a). Automated profile classification was used to separate the high vertical resolution (2.5 mm) estimates of snow density and produce layer-type distributions for rounded (b), faceted (c)m and depth hoar (d) classes.**

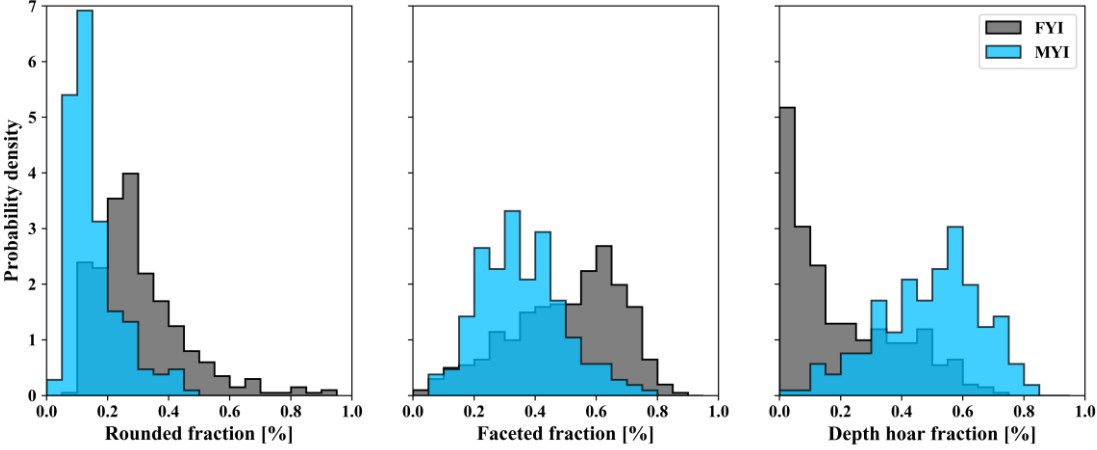

**Figure 8: Fractional snowpack composition by rounded, faceted, and depth hoar layers from the SMP profiles on first year (FYI) and multiyear (MYI) sea ice. Classification methods for the SMP are described in Sect. 3.4.**




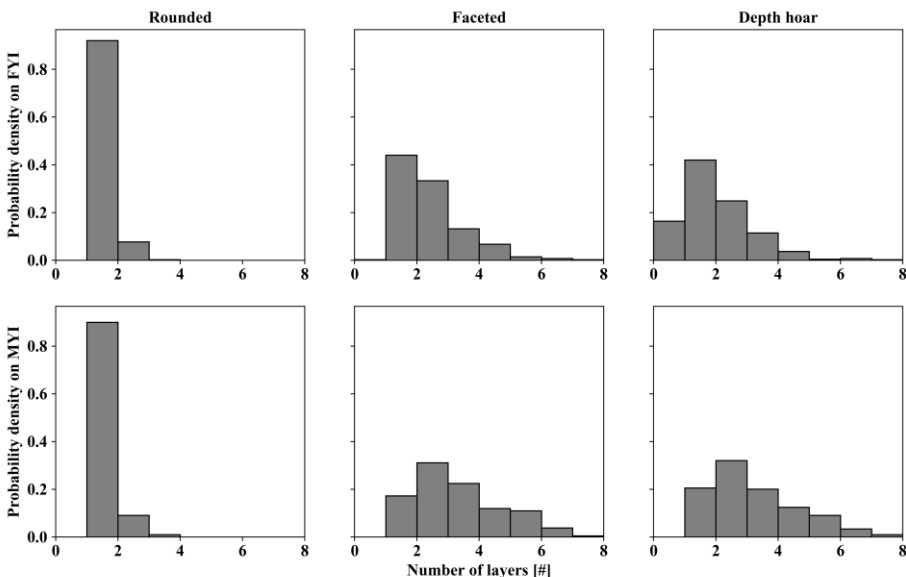

**Figure 9: Distribution of the number of snowpack layers as derived from the SMP transect profiles. Transitions between layer-types from the automated SMP profile classifications are used as a proxy for traditional snow pit layer counts.**


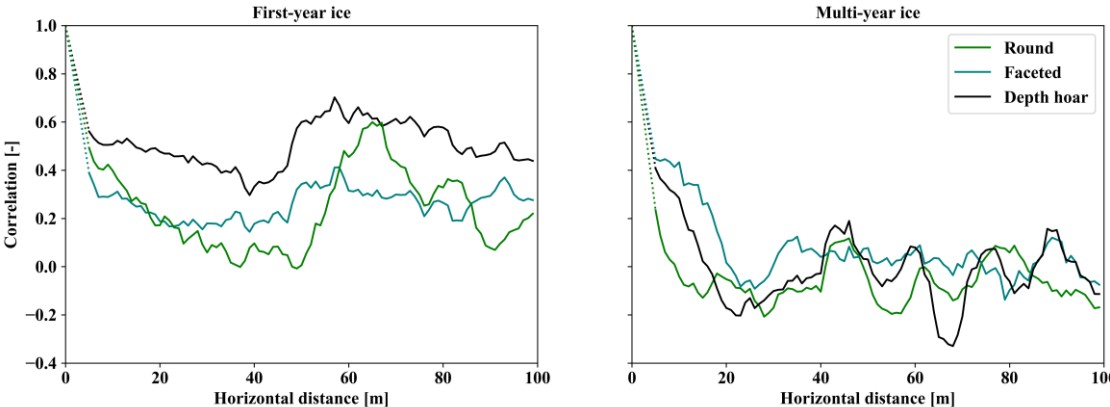

**Figure 10: Spatial auto-correlation of snowpack fractional composition by layer-type on FYI and MYI as estimated from classified SMP profiles. Dotted lines show assumed correlation at length scales less than 1 m where geolocation uncertainty of the profiles**
**precludes analysis.**


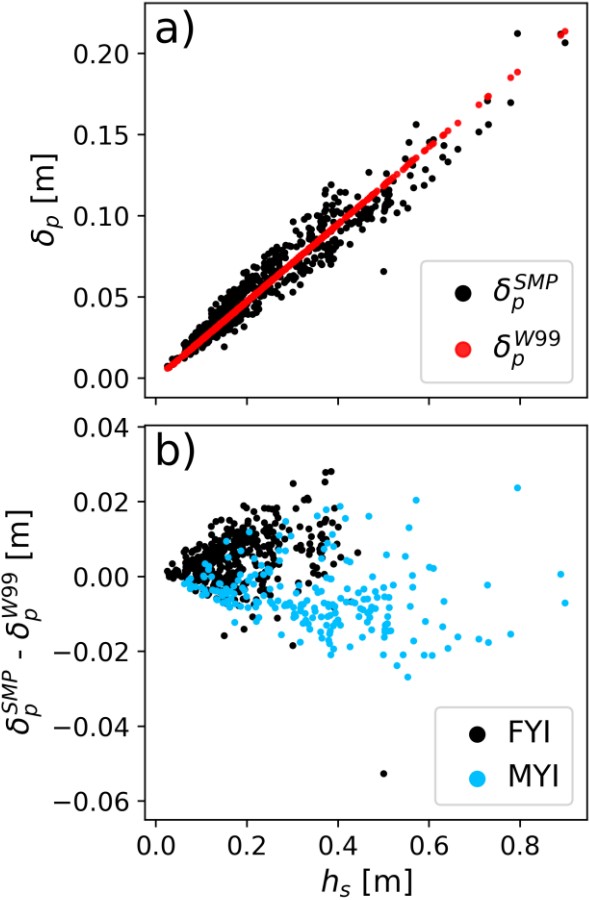

**Figure 11:** Changes in estimated radar propagation bias ($\delta_p$; a) relative to snow thickness ($h_s$) based on density estimated from climatology ($\delta_p^{w99}$) and measured from SMP profiles ($\delta_p^{SMP}$). The two sets of estimates were subtracted to show potential errors associated with the use of climatology over known snow densities (b).