# Peer review of "Local-scale variability of snow density on Arctic sea ice"

_The Cryosphere, 2019_

## Referee Comment (RC1) · Anonymous Referee #1 · 5 Apr 2020

General comments: This paper presents the snow pit and SnowMicroPen measurements over sea ice to recalibrate the SMP density model. The calibrated snow density and machine learning-based layer classification are combined to estimate density and length scale of variability differences in the composition of snow layers. Such density model and data are highly valuable in sea ice altimetry application as mentioned by authors. This in situ and model work are important in snowpack properties analysis and will draw wide interests from the community. This article is well-written and easy to follow. My major comments are as follows: Section 3.2 about how to estimate density form SMP profile is not quite clear to me in P6, L168 'Estimates of _smp were then extracted ...'. From my understanding, what you are doing here is more like getting the original 5cm-thickness _smp profile scaled according to perturbed thickness inn

individual layer. What do you mean by "average the scaled profiles within 3-cm height of cutter measurements"? What does it mean by: "Another 6cm window moving averaging"? P6, L180 and Figure 4, when you compare the density, do you compare each layer mean snow pit density and all SMP profiles estimation at that layer in one site? I noticed that in Eureka, one site has 2 or 3 pits (the distances between these pits are under 100m), how to divide the SMP measurements for these pits if SMPs have the same distance between two sites? In section 3.4, when you use SVM to classify the snow layer's type, with 75% accuracy, have you tried other machine learning methods and have you tried other non-linear kernels except for the linear one? What is the accuracy in other methods, and what are the potential limitation of such methods in classifying snow properties? P7, L219, what is the vertical resolution when snow pit and SMP measurements are both trained considering their vertical resolutions are different. Also, I am very curious about the results when further adding ice type information in the training. P9, L268, 'Profiles collected on FYI, and therefore exclusively near Eureka...'. Do you mean in Figure 7(a), over FYI, the distribution is negatively skewed? But from the figures, the density seems positively skewed over FYI. Also, the following sentence 'In contrast, densities on MYI were positively skewed...'. Please check it. P9, L277, 'Measurements classified as faceted had on average a density...'. Figure 8c is over depth hoar not faceted and the distribution is not negatively skewed. How to quantify the density uncertainty/error from the SMP density model in consideration of application on altimetry studies?

Specific comments: P2, L50, 'Laxon et al. 2013' should be 'Laxon et al., 2013' P9, L271, 'However, these difference...' should be 'However, these differences' P13, L381, 'however the errors appears' should be 'however the errors appear'

---

## Referee Comment (RC2) · Anonymous Referee #2 · 8 Jun 2020

**1 Summary**

This manuscript addresses the spatial variation in the density of snow on sea ice through use of an extensive in-situ dataset from SMP and density-cutters. The paper is well written and highly rigorous; I believe it makes a significant contribution to the study of snow on sea ice and I recommend it for publication in The Cryosphere after some minor changes and clarifications.

On a side note, it was particularly pleasing to see the authors publishing their data and analysis code in an interactive, browser-based environment. As well as making the research output easier to review, it is likely to add to the impact of the work.

**2 Minor Changes and Suggestions**

- L57: "were used **to** address this problem"

- L58/339/471: Perhaps 'mm-scale' should be replaced with 'milimeter-scale' for readability.

- L129: "SMP transects **were** established"

- L184: "Eureka had a higher RMSE ... **than** measurements at AO sites"

- You'll presumably update your coefficient names to reflect the year of publication (K19a → K20a) in the final copy.

- L223: I think the reporting of the classifier's accuracy evaluation could be reworded for clarity. Presumably the 'prediction accuracy of 76%' means that 76% of the samples were assigned the correct layer type? Or does it mean that of the bulk layers that it identified (e.g. depth hoar, slab etc), they were right 76% of the time? Since the SMP makes measurements of F & L a couple of hundred times per mm, then does your classifier make a classification of the snow type with similar frequency, or is it as the frequency of your 2.5 mm density estimates? Or does it just identify boundaries between layers of different snow type?

- I think it would also be particularly valuable to break down the performance by layer-type. The average was 76%, but did the classifier do a better job of identifying different types? Were there some types that were particularly hard to identify?

- L326: As you subsequently mention, the primary scattering surface for radar altimetry may not be the ice surface. As such, I think this should be rephrased as 'radar measured distance to the primary scattering horizon may be overestimated'. On that note, I think you should mention explicitly in this section that calculations of $\delta_p$ assume (in line with convention for radar altimetry) that the ice surface is the dominant scattering horizon.

- L327: This reference is now quite challenging for many readers to track down, I suggest updating to the more recent edition: Ulaby and Long (2014).

- L332: I think it would be good to cite this equation (as it's reported differently in some literature), consider Tilling et al. (2018) or Mallett et al. (2020).

- L444: Consider pointing out in this section that as well as brine over FYI, morphological features in the snow or higher snow temperatures (Willatt et al., 2011) may also raise the primary scattering horizon, limiting the applicability of your path difference calculation.

**References**

Mallett, R. D., Lawrence, I. R., Stroeve, J. C., Landy, J. C., and Tsamados, M.: Brief communication: Conventional assumptions involving the speed of radar waves in snow introduce systematic underestimates to sea ice thickness and seasonal growth rate estimates, Cryosphere, 14, 251–260, https://doi.org/10.5194/tc-14-251-2020, 2020.

Tilling, R. L., Ridout, A., and Shepherd, A.: Estimating Arctic sea ice thickness and volume using CryoSat-2 radar altimeter data, Advances in Space Research, 62, 1203–1225, https://doi.org/10.1016/j.asr.2017.10.051, URL https://doi.org/10.1016/j.asr.2017.10.051, 2018.

Ulaby, F. and Long, D.: Microwave Radar and Radiometric Remote Sensing, The University of Michigan Press, https://doi.org/10.3998/0472119356, URL http://public.ebookcentral.proquest.com/choice/publicfullrecord.aspx?p=4537961, 2014.

Willatt, R., Laxon, S., Giles, K., Cullen, R., Haas, C., and Helm, V.: Ku-band radar penetration into snow cover on Arctic sea ice using airborne data, Annals of Glaciology, 52, 197–205, https://doi.org/10.3189/172756411795931589, 2011.

---

## Author Comment (AC1) · 17 Jul 2020

*General comments: This paper presents the snow pit and SnowMicroPen measurements over sea ice to recalibrate the SMP density model. The calibrated snow density and machine learning-based layer classification are combined to estimate density and length scale of variability differences in the composition of snow layers. Such density model and data are highly valuable in sea ice altimetry application as mentioned by authors. This in situ and model work are important in snowpack properties analysis and will draw wide interests from the community. This article is well-written and easy to follow.*

**Thank you for your review and helpful suggestions to improve the paper. We have made changes throughout section 3.2 to improve our description of how the SMP and density cutter measurements were compared. We have also revised some of the statistical descriptions as suggested. Inline responses to suggestions and questions are provided in bold below.**

*My major comments are as follows: Section 3.2 about how to estimate density form SMP profile is not quite clear to me in P6, L168 'Estimates of _smp were then extracted...'. From my understanding, what you are doing here is more like getting the original 5cm-thickness _smp profile scaled according to perturbed thickness inn individual layer. What do you mean by "average the scaled profiles within 3-cm height of cutter measurements"?*

**Once the SMP profiles are scaled we simply take the corresponding SMP values at the same height of each density cutter measurement. Because the density cutter is 3-cm in height we average the much higher resolution SMP estimates to make a 1:1 comparison. There is scope in the future to optimize how this comparison is made but we have not completed an extensive evaluation here. We modified the sentence on line L169 to make clear what is being averaged.**

*What does it mean by: "Another 6cm window moving averaging"?*

**We hope that the above response clarifies how the matching process was applied. However, we have not discussed a 6 cm moving average and are unsure which lines this comment is referring to.**

*P6, L180 and Figure 4, when you compare the density, do you compare each layer mean snow pit density and all SMP profiles estimation at that layer in one site?*

**Comparisons described here were between each density cutter measurement and the mean of the SMP estimates within their corresponding 3-cm height. Effectively each point in Figure 4 represents a single density cutter measurement. We've made small improvements to the text in an attempt to make this clear.**

*I noticed that in Eureka, one site has 2 or 3 pits (the distances between these pits are under 100m), how to divide the SMP measurements for these pits if SMP shave the same distance between two sites?*

**This is correct, not all sites have the same number of pits, and at times, they are unequally spaced. Placement of the snow pits was structured to characterize inter-site variability but the distance between each was not considered as part of our analysis. All analysis of spatial variability used the**

**distance between SMP profiles (GPS located). We relied on large SMP data volumes rather than strict spacing of profiles to understand scales between 0 and 100 m.**

*In section 3.4, when you use SVM to classify the snow layer's type, with 75% accuracy, have you tried other machine learning methods and have you tried other non-linear kernels except for the linear one? What is the accuracy in other methods, and what are the potential limitation of such methods in classifying snow properties?*

**Thank you for highlighting this important area of future work. We chose to apply a linear kernel with the SVM to limit complexity and focus on broader aspects of the density analysis. There are certainly non-linear divisions within the parameter space which the hyperplanes fail to delineate, limiting accuracy. To apply a non-linear kernel would require an extensive evaluation of the hyper-parameters which we feel is beyond the scope of this work. The work of Havens et al., (2012) stands as a strong example that enhanced SMP classification methods can be applied to improve accuracy. We hope to conduct an extensive assessment of other classification methods in future work.**

*P7, L219, what is the vertical resolution when snowpit and SMP measurements are both trained considering their vertical resolutions are different. Also, I am very curious about the results when further adding ice type information in the training.*

**Adding ice type information resulted in a small improvement of ~2% accuracy. The example we created will remain in our revised public code for reference. We chose not to use this configuration as ice type as ancillary information is not directly available from the SMP.**

*P9, L268, 'Profiles collected on FYI, and therefore exclusively near Eureka...'. Do you mean in Figure 7(a), over FYI, the distribution is negatively skewed? But from the figures, the density seems positively skewed over FYI. Also, the following sentence 'In contrast, densities on MYI were positively skewed...'. Please check it.*

**We have revised wording throughout the paper to use left- or right-skewed instead of negative or positive. The distribution in question is now described as left-skewed (Statistical skew of -0.41).**

P9, L277, 'Measurements classified as faceted had on average a density...'.Figure 8c is over depth hoar not faceted and the distribution is not negatively skewed.

**Thank you for noting the incorrect label, we have corrected this. See our previous comment regarding skew.**

*How to quantify the density uncertainty/error from the SMP density model in consideration of application on altimetry studies?*

**Errors quantified in the study showed the SMP to be comparable to those expected from manual density cutter measurements. We hope to use this information to build a more comprehensive**

analysis of errors involved in altimetry of sea ice. However, we do not have any specific conclusions at this point on how best to address uncertainty when applying the SMP to altimetry studies.

*Specific comments: P2, L50, 'Laxon et al. 2013' should be 'Laxon et al., 2013' P9,L271, 'However, these difference...' should be 'However, these differences' P13, L381,'however the errors appears' should be 'however the errors appear'*

**Thank you for noting these errors. Each has been revised as suggested.**

---

## Author Comment (AC2) · 17 Jul 2020

*This manuscript addresses the spatial variation in the density of snow on sea ice through use of an extensive in-situ dataset from SMP and density-cutters. The paper is well written and highly rigorous; I believe it makes a significant contribution to the study of snow on sea ice and I recommend it for publication in The Cryosphere after some minor changes and clarifications.*

*On a side note, it was particularly pleasing to see the authors publishing their data and analysis code in an interactive, browser-based environment. As well as making the research output easier to review, it is likely to add to the impact of the work.*

**Thank you for your review and comments to improve the quality of the manuscript. We sincerely hope that the methods and code presented here can be made better through community application and adaptation. Thank you for your note regarding our efforts to ensure reproducible and open science.**

*L57: "were used to address this problem"*

**Corrected as suggested.**

*L58/339/471: Perhaps 'mm-scale' should be replaced with 'milimeter-scale' for readability.*

**We have made the units explicit as suggested.**

L129: "SMP transects **were** established"

**Corrected as suggested.**

*L184: "Eureka had a higher RMSE ...**than** measurements at AO sites"*

**Corrected as suggested.**

*You'll presumably update your coefficient names to reflect the year of publication (K19a→K20a) in the final copy.*

**Updated throughout to reflect the current year.**

L223: I think the reporting of the classifier's accuracy evaluation could be reworded for clarity. Presumably the 'prediction accuracy of 76%' means that 76% of the samples were assigned the correct layer type? Or does it mean that of the bulk layers that it identified (e.g. depth hoar, slab etc), they were right 76% of the time?

**Added text to clarify what accuracy means (True positive or true negative predictions on line 224). We have also added an evaluation of errors by layer type and reported classification errors as a confusion matric (Table 3).**

Since the SMP makes measurements of F & L a couple of hundred times per mm, then does your classifier make a classification of the snow type with similar frequency, or is it as the frequency of your 2.5 mm density estimates? Or does it just identify boundaries between layers of different snow type?

**The classifier is trained on the 3-cm averaged SMP data extracted at the height of each density cutter measurement. We added text to make it clear that the classifier was applied at vertical resolution of the SMP despite training on density cutter averages.**

I think it would also be particularly valuable to break down the performance by layer-type. The average was 76%, but did the classifier do a better job of identifying different types? Were there some types that were particularly hard to identify?

**Agreed, see previous response regarding the introduction of the confusion matrix.**

L326: As you subsequently mention, the primary scattering surface for radar altimetry may not be the ice surface. As such, I think this should be rephrased as 'radar measured distance to the primary scattering horizon may be overestimated'. On that note, I think you should mention explicitly in this section that calculations of $\delta p$ assume (in line with convention for radar altimetry) that the ice surface is the dominant scattering horizon.•

**Agreed, more careful wording was needed to acknowledge penetration uncertainty. We've modified the text to establish that our assumption is that the primary scattering interface is the ice surface. See further points below where additional details have been added on why that assumption may be invalid.**

L327: This reference is now quite challenging for many readers to track down, I suggest updating to the more recent edition: Ulaby and Long (2014).•

**Thanks for your suggestion but we have kept the original given differences in authorship.**

L332: I think it would be good to cite this equation (as it's reported differently in some literature), consider Tilling et al. (2018) or Mallett et al. (2020).

**A reference to Tilling et al. (2018) has been added as suggested to establish linage.**

L444: Consider pointing out in this section that as well as brine over FYI, morphological features in the snow or higher snow temperatures (Willatt et al., 2011) may also raise the primary scattering horizon, limiting the applicability of your path difference calculation.

**Yes, important to mention this even if assumed. We added a point to clarify that snow conditions were dry and temperature was not considered. We will leave the final sentence as is using microstructure to address the influence of layering.**